

# Changes in the Width of the Tropical Belt due to Simple Radiative Forcing Changes in the GeoMIP Simulations

Nicholas Davis[1], Dian J. Seidel[2], Thomas Birner[1], Sean M. Davis[3], and
Simone Tilmes[4]

[1]Department of Atmospheric Science, Colorado State University, Fort Collins, CO
[2]NOAA Air Resources Laboratory, College Park, MD (retired)
[3]NOAA Earth System Research Laboratory, Boulder, CO
[4]National Center for Atmospheric Research, Boulder, CO

*Correspondence to:* Nicholas Davis (nadavis@atmos.colostate.edu)

**Abstract.** Model simulations of future climates predict a poleward expansion of subtropical arid climates at the edges of earth's tropical belt, which would have significant environmental and societal impacts. This expansion may be related to the poleward shift of the Hadley cell edges, where subsidence stabilizes the atmosphere and suppresses precipitation. Understanding the primary drivers

of tropical expansion is hampered by the myriad forcing agents in most model projections of future climate. While many previous studies have examined the response of idealized models to simplified climate forcings and the response of comprehensive climate models to more complex climate forcings, none have examined how comprehensive climate models respond to simplified climate forcings. To shed light on robust processes associated with tropical expansion, here we examine how

the tropical belt width, as measured by the Hadley cell edges, responds to simplified forcings in the Geoengineering Model Intercomparison Project (GeoMIP). The tropical belt expands in response to a quadrupling of atmospheric carbon dioxide concentrations and contracts in response to a reduction in the solar constant, with a range of a factor of three in the response among nine models. Models with more surface warming and an overall stronger temperature response to quadrupled carbon dioxide

ide exhibit greater tropical expansion, a robust result in spite of intermodel differences in the mean Hadley cell width, parameterizations, and numerical schemes. Under a scenario where the solar constant is reduced to offset an instantaneous quadrupling of carbon dioxide, the Hadley cells remain at their preindustrial width, despite the residual stratospheric cooling associated with elevated carbon dioxide levels. Quadrupled carbon dioxide produces greater tropical belt expansion in the Southern

Hemisphere than in the Northern Hemisphere. This expansion is strongest in austral summer and autumn. Ozone depletion has been argued to cause this pattern of changes in observations and model experiments, but the results here indicate that seasonally- and hemispherically-asymmetric tropical expansion can be a basic response of the general circulation to climate forcings.





## 1   Introduction

Earth's tropical belt can be defined by the band of rainy equatorial regions bordered by the arid
subtropics to the north and the south. The Hadley cells, two thermally-direct tropospheric circula-
tions with rising motion near the equator, significantly influence the surface climate of the tropical
belt. Converging easterly near-surface trade winds transport moisture into the Intertropical Conver-
gence Zone, a meandering front of convection that brings rain to the equatorial latitudes and heats

tropical air through the condensation of water vapor. This heated air rises through the troposphere
and diverges poleward into the upper troposphere of both hemispheres, eventually subsiding in the
subtropics where it dries and stabilizes the atmosphere against convection. Because of the strong
latitudinal gradients in temperature and precipitation at the edges of the tropical belt, any shift in its
edges could drive major changes in surface climate (Birner et al., 2014).

There is mounting evidence that such changes are already taking place. Soil moisture (Dorigo
et al., 2012), precipitation (New et al., 2001; Zhang et al., 2007), and sea surface salinity (Helm et al.,
2010) trends over the past several decades are consistently indicate an intensification and poleward
shift of the hydrological cycle. The intensification is widely considered to be driven primarily by
increasing water vapor concentrations in a warming atmosphere (Held and Soden, 2006), but the

circulation changes that drive poleward shifts in the hydrological cycle are not as well understood.
Further subtropical drying and a poleward expansion of arid lands is projected to continue (Lu et al.,
2007; Scheff and Frierson, 2012; Feng and Fu, 2013).

Evidence of tropical expansion has been reported based on satellite observations of outgoing long-
wave radiation (Hu and Fu, 2007; Johanson and Fu, 2009; Hu et al., 2011; Fu and Lin, 2011) and

total column ozone (Hudson et al., 2003; Hudson, 2012). Observational estimates of the tropical belt
width based on dynamical fields, such as the subtropical ridges in sea level pressure, also indicate
tropical expansion, although the trends are weaker than those based on outgoing longwave radiation
and precipitation metrics (Hu et al., 2011).

Other metrics for the tropical belt edge latitudes, such as the latitudes of the jet streams (Archer

and Caldeira, 2008; Fu and Lin, 2011; Davis and Birner, 2013) and the latitudes of the subtropical
tropopause breaks (Seidel and Randel, 2007; Birner, 2010; Davis and Rosenlof, 2012; Lucas et al.,
2012; Davis and Birner, 2013; Ao and Hajj, 2013; Lucas and Nguyen, 2015) indicate historical
tropical expansion, as well. An expansion of the Hadley cells has been detected in reanalyses (Hu
and Fu, 2007; Johanson and Fu, 2009; Stachnik and Schumacher, 2011; Davis and Rosenlof, 2012;

Davis and Birner, 2013; Nguyen et al., 2013; Chen et al., 2014). Tropical expansion estimates based
on reanalyses, however, may suffer from spurious trends and discontinuities in basic meteorological
fields (Trenberth et al., 2001; Bengtsson et al., 2004). The rate of Hadley cell expansion and even the
mean strength of the Hadley cells varies among the reanalyses (Stachnik and Schumacher, 2011),
which could indicate that the meridional winds are not well constrained. There is also significant

uncertainty in the observed rate of tropical expansion because it is highly variable for different





metrics and data products (Birner, 2010; Davis and Rosenlof, 2012; Davis and Birner, 2013; Lucas et al., 2014).

Attributing surface impacts to tropical expansion and attributing tropical expansion itself to particular climate forcings is difficult given the number of external forcings changing over the historical period, as well as the impact of natural climate variability on the trends. Factors such as the Pacific Decadal Oscillation, the El Niño-Southern Oscillation (Lu et al., 2008), and the Southern Annular Mode influence the tropical belt width and may explain non-negligible fractions of its historical trend (Grassi et al., 2012; Allen et al., 2014; Lucas and Nguyen, 2015; Garfinkel et al., 2015).

Climate model simulations offer an avenue for assessing the response of the Hadley cells and tropical belt to different climate forcings and forcing evolutions, and long integrations minimize the impact of interannual variability (Hawkins and Sutton, 2009). Both Lu et al. (2009) and Hu et al. (2013) found that significant tropical expansion occurs only when greenhouse gas concentrations increase in historical climate simulations. Increasing greenhouse gas concentrations in future climate simulations similarly cause the tropical belt to expand relative to its preindustrial control width (Gastineau et al., 2008), with the amount of expansion scaling with the concentration of greenhouse gases (Lu et al., 2007; Tao et al., 2015). However, Adam et al. (2014) have shown that the Hadley cell width is generally sensitive to changes in both mean sea surface temperatures and meridional temperature gradients. Any climate forcing that modifies mean temperatures or their gradients could thus drive variations in the tropical belt width. Stratospheric ozone depletion and its resulting polar stratospheric cooling has been argued to be a potentially dominant driver of Southern Hemisphere tropical expansion (Polvani et al., 2011b; Min and Son, 2013), and ozone recovery over the coming decades may oppose any future greenhouse-gas-driven expansion (Son et al., 2009; Polvani et al., 2011a). Black carbon, tropospheric ozone (Allen et al., 2012), and aerosols (Allen and Sherwood, 2011; Allen et al., 2014) may have also played a role in historical tropical expansion, especially in the Northern Hemisphere. While examining the response of climate models to realistic sets of past and future forcings is appealing, it is not ideal for identifying how the tropical belt responds to particular forcings. Many climate forcing agents are simultaneously changing in these simulations, and separating their effects is often intractable.

Idealized modeling, which involves changing a single climate forcing or model parameter, complements those more realistic simulations. The models are often simplified versions of fully-coupled climate models that may solve only the equations of motion and thermodynamics without explicitly resolving radiation and convection. Polvani and Kushner (2002) and Kushner and Polvani (2004) found that stratospheric cooling in such an idealized model produced a poleward shift of the midlatitude jet. It also produced a poleward shift in the pattern of surface easterlies and westerlies which indicates an expansion of the tropical belt. While Lorenz and DeWeaver (2007) found that cooling the stratosphere by raising the height of the tropopause was sufficient to produce a poleward shift of the tropospheric jets, Tandon et al. (2011) found that stratospheric cooling without perturbing





the tropopause height was sufficient to drive an expansion of the Hadley cells. Similar to Tandon et al. (2011), Maycock et al. (2013) found that idealized increases in stratospheric water vapor drove

enhanced stratospheric cooling and a poleward shift of the tropospheric jets. In the troposphere, tropical and subtropical warming can also drive an expansion of the Hadley cells (Frierson et al., 2007; Tandon et al., 2013). Thus, stratospheric cooling, tropospheric warming, and increasing the height of the tropopause can all independently drive poleward shifts in the circulation.

However, idealized models do not explicitly model clouds or cloud-related feedbacks. Convection

is a fundamental aspect of the Hadley cells (Frierson, 2007), and cloud radiative effects can impact modeled circulation changes (Ceppi et al., 2012, 2014; Voigt and Shaw, 2015). What is lacking is a study that applies simple climate forcings in cleanly-designed experiments to fully-coupled models to bridge the gap between the existing idealized and more realistic model simulations.

In this study, we will examine the response of the tropical belt to different forcings in the Geo-

engineering Model Intercomparison Project (GeoMIP) (Kravitz et al., 2011). GeoMIP, a companion project to the Coupled Model Intercomparison Project Phase 5 (CMIP5) (Taylor et al., 2012), is designed to improve the understanding of the response of the earth system to idealizations of different proposed climate geoengineering activities. Geoengineering impacts aside, the GeoMIP experiments offer a unique opportunity to study the response of fully-coupled climate models to very simple cli-

mate forcings, which may shed light on the processes responsible for observed past and possible future tropical width changes.

## 2   Data and methods

While numerous climate forcings can impact the width of the tropical belt, we focus on variations in carbon dioxide and insolation simulated in GeoMIP. Our analysis is based on monthly-mean output

from nine climate models (Table 1) that performed three sets of experiments: the GeoMIP Geoengineering 1 (G1) experiment (Kravitz et al., 2011), the preindustrial control (piControl), and the abrupt quadrupled carbon dioxide ($4 \times CO_2$) experiments in CMIP5 (Taylor et al., 2012). The piControl experiment fixes all climate forcings at preindustrial levels to provide an estimate of the unperturbed climate system and will be the control experiment in this study. The $4 \times CO_2$ experiment applies

an instantaneous quadrupling of piControl carbon dioxide concentrations, while the G1 experiment balances this abrupt quadrupling with a decrease in the solar constant such that the global-mean top-of-atmosphere radiative forcing is zero (Kravitz et al., 2011). This crudely models the effect of a global climate intervention scheme based on albedo modification (National Research Council, 2015), but more generally tests the impact of a decrease in insolation on the climate system, with

some relevance for paleoclimate research. We only use the G1 experiment from GeoMIP because of its simple forcing scheme that is applied uniformly in all models.





For the G1 experiment, not all models achieved a perfect cancellation of the top-of-atmosphere radiative forcings. Table 1 lists the top-of-atmosphere radiative forcing in the $4\times CO_2$ experiment and the residual top-of-atmosphere radiative forcing in the G1 experiment after the solar constant reduction for each model (e.g., Huneeus et al. (2014)).

Because the $4\times CO_2$ and G1 experiments involve an abrupt forcing at the start of the simulation, we discard the first 5 years of each experiment, a conservative choice as the circulation metrics adjust to the abrupt forcing within two years. The piControl simulations from each model range from 500 to 3000 model years, the $4\times CO_2$ simulations range from 140 to 150 model years, and the G1 simulations range from 50 to 100 model years. For each experiment, we use the same number of model years from each model simulation based on the shortest simulation, e.g., for the piControl experiment we use the first 500 years from all of the model simulations.

All calculations and analyses use monthly-mean model output. For testing the significance of changes in the tropical belt edge latitudes and width we use two-sided Student's t-tests for the difference of means with unequal variances and sample sizes. The tests thus take into account the different lengths and internal variability of each experiment. We use the effective degrees of freedom, which are are calculated using the lag-1 autocorrelation of the monthly-mean anomalies (Bretherton et al., 1999). Differences are deemed statistically significant for $p \leq 0.05$ (the 95% confidence level).

## 2.1 Tropical belt edge metric

We define the tropical belt edge latitudes as the latitudes where the vertically-averaged mean meridional streamfunction is zero, poleward of its tropical maximum (minimum) in the Northern (Southern) Hemisphere (Davis and Birner, 2013). The tropical belt width is defined as the difference, in degrees latitude, between the Northern and Southern Hemisphere edge latitudes. The mean meridional streamfunction is the vertical integral of the zonal-mean meridional mass flux between a given level and the top of the atmosphere, and is the primary field used to study variations in the Hadley cells' width and intensity. It is expressed mathematically as

$$\Psi(p,\phi) = \frac{2\pi a \cos(\phi)}{g} \int\limits_{p}^{0} [v] dp \tag{1}$$

were $\Psi$ is the mean meridional streamfunction at the pressure $p$ and latitude $\phi$, $[v]$ is the zonal-mean meridional wind, $a = 6.371 \times 10^6$ m is the mean radius of the earth, and $g = 9.81$ ms$^{-2}$ is the acceleration due to gravity. While the Hadley cell edge latitudes are often calculated as the latitudes where the 500 hPa streamfunction is zero, the choice of a single, arbitrary pressure level subjects the metric to spurious trends due to mean-state changes, such as a deepening of the troposphere, and to intermodel differences in this circulation (Birner, 2010; Davis and Rosenlof, 2012; Davis and Birner, 2013). Instead we vertically-average the streamfunction in pressure before calculating the edge latitudes. The interpretation of this vertical average of the streamfunction is simple: it measures





the average meridional overturning circulation strength at a given latitude, and the latitude where it is zero indicates the separation of the Hadley and Ferrel cells.

We note that this metric and our analyses focus on the zonal mean. However, historical tropical expansion exhibits significant zonal asymmetries (Chen et al., 2014; Lucas and Nguyen, 2015),
and some zonally asymmetric dynamics contribute to the longitudinal structure of the meridional overturning circulation (Karnauskas and Ummenhofer, 2014).

### 2.2 Tropical belt edge locations

Before analyzing the $4\times CO_2$ and G1 experiments, we will first examine the climatology of the tropical belt edge latitudes in the piControl experiment (Fig. 1). The median tropical belt edge latitudes in
each hemisphere are comparable among the models. In general, models with more equatorward edge latitudes in one hemisphere have more equatorward edge latitudes in the other hemisphere. There is greater interannual variability in the Northern Hemisphere edge latitude, which is borne out in reanalyses and observations (Davis and Birner, 2013). Some models, including the IPSL-CM5A-LR and CSIRO-Mk3L-1-2 models, have little interannual variability in their Southern Hemisphere edge
latitudes.

### 3 Temperature response

We will first characterize the temperature changes in each model between the $4\times CO_2$ and piControl and between the G1 and piControl experiments. The motivation to examine the basic zonal-mean temperature response in all nine models is threefold: (1) temperature changes are associated with
changes in the tropical belt width (e.g., Adam et al. (2014)), (2) the zonal-mean temperature response may provide information about a model's sensitivity to different forcings, and (3) examining only the multi-model-mean may obscure important information about the robustness of the response and its intermodel variations.

Quadrupled carbon dioxide concentrations drive the expected surface and tropospheric warming
and stratospheric cooling Manabe and Wetherald (1967) (Fig. 2). The tropical upper-tropospheric warming is due to moist adiabatic adjustment communicating the surface warming to upper levels (Held et al., 1993; Romps, 2011). Enhanced Arctic warming, or "Arctic amplification", is partly due to decreases in surface albedo brought on by reductions in snow cover and sea ice (Pithan and Mauritsen, 2014) and enhanced downwelling longwave radiation through the so-called "ice-insulation"
feedback (Burt et al., 2015). The stratospheric cooling is driven primarily by enhanced infrared cooling to space due to increased carbon dioxide concentrations. However, other processes may contribute to the cooling as its spatial structure is far from uniform. While all models capture this canonical greenhouse gas response in zonal-mean temperature, the temperature changes vary by nearly a factor of three. The IPSL-CM5A-LR has the strongest response with 13 K upper-tropospheric and





Arctic warming, while the CCSM4 model has the weakest response with 5 K upper-tropospheric and
8 K Arctic warming.

The G1 experiment's solar constant reduction generally balances most of the warming from
quadrupled carbon dioxide (Fig. 3). Because Fig. 3 shows the difference in temperature between
the G1 and piControl experiments, it can be interpreted as the temperature response to $4\times CO_2$ that
is *not* counteracted by the solar constant reduction in the G1 experiment. In the G1 experiment, the
stratosphere is cooler than it is in the piControl experiment in all models because of the reduction in
absorbed solar radiation and infrared radiation emission by the (still enhanced) carbon dioxide con-
centrations. However, the troposphere is marginally cooler in some models (CCSM4, GISS-E2-R,
and MIROC-ESM) and marginally warmer in others (CanESM2, HadGEM2-ES, and MPI-ESM-
LR). Unlike the robust temperature response in the $4\times CO_2$ experiment, there is no robust residual
warming or cooling in the troposphere in G1 compared to piControl. Contrary to expectations, the
model with the strongest residual radiative forcing in the G1 experiment, GISS-E2-R, does not have
a warmer troposphere, while one of the models with a radiative forcing of zero, CanESM2, has
a significantly warmer troposphere. In the coming sections, we will explore how the tropical belt
responds to these simple forcings and whether any processes could explain such changes.

## 4 Tropical belt width response

Quadrupled carbon dioxide drives a statistically significant expansion of the tropical belt as measured
by the Hadley cell edge latitudes in both the Southern and Northern Hemisphere (Fig. 4). There is
a large spread in the magnitude of tropical expansion, though, with values ranging from 1 degree of
total (width) expansion in the CSIRO-Mk3L-1-2 model to nearly 7 degrees of total expansion in the
IPSL-CM5A-LR model (the model with the strongest temperature response to quadrupled carbon
dioxide). The nearly factor of seven difference in the circulation response is far larger than the factor
of three temperature response difference.

More surprising is that the Southern Hemisphere expansion is on average twice the Northern
Hemisphere expansion (Fig. 4). Southern Hemisphere stratospheric ozone depletion has been ar-
gued to be a dominant driver of the more rapid observed expansion of the Southern Hemisphere
Hadley cell (Polvani et al., 2011b; Min and Son, 2013; Waugh et al., 2015). However, the results
here indicate that even with a hemispherically-symmetric climate forcing which does not include
ozone changes, the tropical belt responds asymmetrically with greater expansion in the Southern
Hemisphere. Furthermore, the expansion is strongest in the Southern Hemisphere in austral summer
and autumn (Fig. 5). These are the seasons when the stratospheric cooling due to ozone depletion is
expected to have its greatest impact on Southern Hemisphere expansion trends as ozone is depleted
throughout austral spring.





The solar constant reduction in the G1 experiment counteracts most of the $CO_2$-driven expansion
in the $4\times CO_2$ experiment, despite the residual stratospheric cooling. This suggests that stratospheric
cooling on the order of 1-6 K with the maximum cooling over the poles is not sufficient to appre-
ciably widen the tropical belt (Fig. 3). However, the altitude of the cooling may be an important
factor in determining whether the tropical belt responds or not. For example, in idealized dry simu-
lations Tandon et al. (2011) found that extratropical stratospheric cooling must extend down to the
tropopause to drive a strong circulation response. In the G1 experiment, the cooling is well above the
typical height of the extratropical tropopause (Fig. 3), which is generally located at approximately
250-300 hPa. This may be why there is no robust tropical expansion in the G1 experiment. Pro-
cesses in fully-coupled models that are not represented in idealized dry simulations, including cloud
and radiation feedbacks, could act to further damp the response of the tropical belt to stratospheric
cooling.

For most models the differences between their G1 and piControl experiment edge latitudes and
width are small, often less than $\pm 0.5$ degrees latitude (with an average difference of zero). Just as
there is no robust tropospheric temperature difference between the G1 and piControl experiments,
there is no robust residual tropical expansion or contraction. These changes are not statistically
significantly correlated with the residual radiative forcings in the G1 experiment.

In the Northern Hemisphere (Fig. 5), tropical expansion in response to increased carbon dioxide
concentrations is approximately constant from December-January-February (DJF) through June-
July-August (JJA), but slightly larger in September-October-November (SON). The enhanced ex-
pansion in boreal autumn is consistent with realistic (Hu et al., 2013; Kang and Lu, 2012) and more
idealized (Kang and Lu, 2012) CMIP5 forcing simulations and with historical reanalyses (Hu and Fu,
2007). While Allen et al. (2012) proposed that the enhanced tropical expansion in Northern Hemi-
sphere summer and autumn was driven by the combined effects of black carbon and tropospheric
ozone, it appears that increased carbon dioxide concentrations alone could also drive some of this
enhanced expansion. As a caveat, however, the tropical belt contracts in some models and seasons
in response to quadrupled carbon dioxide concentrations, and the spread in the residual tropical belt
width changes between the piControl and G1 experiments is as large as the spread in the tropical
expansion between the $4\times CO_2$ and piControl experiments. This lack of robustness indicates some
uncertainty in the seasonality of Northern Hemisphere tropical expansion in response to increases in
carbon dioxide.

To explore whether the large range in the responses and the asymmetric response in the two hemi-
spheres are associated with any particular temperature structures, we composite the difference in
temperature between the $4\times CO_2$ and piControl experiments in the four models with the greatest and
in the four models with the least tropical expansion (Fig. 6). Both groups show the same general
pattern of tropospheric warming and stratospheric cooling. In fact, the difference in the temperature
response to quadrupled carbon dioxide between the models with the greatest and the least tropical





expansion itself resembles the temperature response to quadrupled carbon dioxide. There are no unique relationships in the strength of the tropical upper-tropospheric amplification, the Arctic amplification, the surface warming, or the stratospheric cooling. Rather, these temperature responses all consistently scale among the models with greater tropical expansion.

### 4.1 Intermodel differences in the tropical width response and associated mechanisms

Subtropical static stability increases due to tropical upper-tropospheric amplification may be important for driving tropical expansion (Fig. 6). Held (2000) derived a scaling theory for the Hadley cell width based on the critical shear for baroclinic instability in the Phillips two-layer model (Phillips, 1951). If one assumes that the poleward flow in the Hadley cells conserves angular momentum, and that the flow terminates at the latitude of the onset of baroclinic instability, then the edge latitude of the Hadley cell is only a function of the tropopause height and the gross static stability (the difference between the potential temperature of the tropopause and the surface). Increases in static stability or tropopause height would both act to further stabilize the flow against baroclinic instability and allow the Hadley cell to expand poleward. Lu et al. (2008) found changes in static stability to be strongly correlated with changes in the Hadley cell edge latitude, and a cursory scale analysis shows that the scaling theory is dominated by the static stability term for typical variations in static stability and tropopause height (Frierson et al., 2007). For these reasons we will focus exclusively on changes in subtropical static stability.

The Held (2000) scaling theory has been successfully used to study tropical expansion in models ranging from dry dynamical cores to fully-coupled climate models (Frierson et al., 2007; Lu et al., 2007, 2008), although modified scaling theories that relax the angular momentum conservation constraint (Kang and Lu, 2012), as well as theories based on other criteria (Lu et al., 2008; Korty and Schneider, 2008; Tandon et al., 2013; Levine and Schneider, 2015) may be more realistic. Similar to Levine and Schneider (2015), we evaluate the gross static stability, hereafter "subtropical static stability", at the tropical belt edge latitude. We define the subtropical static stability as the difference in potential temperature between 100 hPa (approximately the tropical tropopause) and 1000 hPa (approximately the surface) averaged over 5 degrees of latitude equatorward of the tropical belt edge latitude for each month in each hemisphere.

In both hemispheres, tropical expansion between the $4\times CO_2$ and piControl experiments is associated with an increase in subtropical static stability, with the increase in stability explaining 29-55% of the intermodel variation in tropical expansion (Fig. 7). This relationship also holds for the tropical expansion and contraction between the G1 and piControl experiments, where changes in static stability explain 42-46% of the total intermodel variation in the tropical belt edge latitudes. These results are noteworthy for two reasons. Firstly, the relationships remain linear both among models with smaller changes in stability and tropical belt width, as well as among models with larger expansion and contraction. Secondly, despite differences in the models' mean edge latitudes and their



parameterizations of convection and other processes, and despite the dearth of physical intermodel relationships (Davis and Birner, 2016), this particular relationship is robust across models and scenarios.

Tropical upper-tropospheric temperatures tend to warm more than surface temperatures due to moist adiabatic adjustment (Held et al., 1993; Romps, 2011). Because the moist adiabatic lapse rate scales with surface temperature, any change in static stability in the tropics and subtropics reflects changes in surface temperature. Accordingly, tropical expansion in both hemispheres also scales with increases in global-mean surface temperature (Fig. 8), explaining 47-49% of the intermodel variation

in tropical expansion between the $4 \times CO_2$ and piControl experiments. Despite being the residual rather than the forced response, increases in global-mean surface temperature also explain 74% of the intermodel variation in tropical expansion in the Southern Hemisphere in the G1 experiment, though less so in the Northern Hemisphere. Compared to the Southern Hemisphere, Northern Hemisphere tropical expansion seems to scale nonlinearly for large increases in global-mean surface temperature,

explaining its weaker linear correlations.

The nonlinearity extends to the change in the tropical belt width relative to changes in global-mean surface temperature, with tropical expansion disproportionately increasing as the global-mean surface temperature increases (Fig. 9). As is the case for the edge latitudes (Fig. 7), the change in the tropical belt width relative to changes in subtropical static stability is more linear but also more

scattered. Here the change in the subtropical static stability is the average of the change in both hemispheres. Despite the nonlinearity, the change in the change in the tropical belt width is better correlated with the change in global-mean surface temperature than with the change in subtropical static stability, explaining 54-79% of the total intermodel variation in the change in the tropical belt width.

We also examined Arctic warming and tropical upper-tropospheric warming separately, as the two may have different impacts on tropical expansion and/or may explain some additional intermodel variation in the tropical belt response. However, both of these indices are well-correlated with the total change in global-mean surface temperature (Fig. 10). Tropical upper-tropospheric temperature changes are well-correlated with the change in global-mean surface temperature across the models

for both the difference between the $4 \times CO_2$ and piControl experiments and the difference between the G1 and piControl experiments. For the Arctic warming, the correlations do not depend upon whether one defines Arctic amplification as the total temperature change at the surface in the Arctic (as is done here) or as the difference between the total temperature change at the surface in the Arctic minus the change in global-mean surface temperature; if one is correlated with global-mean surface

temperature, the other will be as well.





## 5   Conclusions

We have examined the response of the tropical belt to simple radiative forcing experiments in the GeoMIP experiments. Quadrupled concentrations of carbon dioxide in the $4 \times CO_2$ experiment produce the canonical temperature response and drive significant tropical expansion in all models. The insolation reduction in the G1 experiment generally counteracts the carbon-dioxide-induced tropospheric warming, but leaves the stratosphere colder than it was in the piControl experiment. The lack of any significant change in the tropical belt width between the G1 and piControl experiments indicates that broad stratospheric cooling alone may not drive tropical expansion, at least when the cooling does not extend down to the tropopause.

The expansion in response to quadrupled carbon dioxide concentrations is greater in the Southern Hemisphere and peaks in austral summer and autumn. Both responses have previously been identified as signatures of Antarctic ozone depletion on observed Southern Hemisphere tropical expansion. They instead appear to comprise the basic response of the circulation to simple hemispherically-symmetric, non-ozone climate forcings. This does not imply that ozone depletion and other climate forcings have not contributed to observed tropical expansion. Rather, it may be that ozone depletion and increased greenhouse gas concentrations have together enhanced the expansion in the Southern Hemisphere and in summer and autumn. The Southern Hemisphere Hadley cell may exist in a different dynamical regime than the Northern Hemisphere cell (Davis and Birner, 2013) due to the Southern Hemisphere cell's strong coupling to the eddy-driven jet (Kang and Polvani, 2011; Ceppi and Hartmann, 2013; Staten and Reichler, 2014). This jet has a more robust poleward shift in response to greenhouse gas increases than its Northern Hemisphere counterparts (Barnes and Polvani, 2013) which enhance Southern Hemisphere tropical expansion. Further, the Hadley cells are more susceptible to the influence of extratropical Rossby waves in summer (Schneider and Bordoni, 2008), which may contribute to the seasonality of the expansion in both hemispheres.

Models with a stronger temperature response to increased carbon dioxide (which includes stronger surface, upper-tropospheric, and Arctic warming and stronger stratospheric cooling) have greater tropical expansion. While tropical expansion scales with increases in both subtropical static stability and global-mean surface temperature, these indices effectively measure the same thermodynamic response because of moist adiabatic adjustment. Increases in global-mean surface temperature can explain up to 79% of the total intermodel variation in tropical expansion because it occurs within the intermodel space of fully-coupled climate models. Different mean states (Kidston and Gerber, 2010), the representation of parameterized processes (Frierson, 2007), the strength of cloud feedbacks (Feldl and Bordoni, 2016), and model design choices such as horizontal resolution (Landu et al., 2014; Lorant and Royer, 2001; Davis and Birner, 2016) can all influence the circulation and its response. Tropical belt width changes are thus part and parcel of global climate change. They are strongly correlated with changes in other key climate features and are not a separate phenomenon.



Tropical expansion could be considered as a robust response of the climate system to increasing greenhouse gas concentrations similar to an acceleration of the hydrological cycle.

How the temperature or static stability changes actually drive tropical expansion is an open question. The scaling theory used here includes some unrealistic assumptions. Angular momentum is not perfectly conserved in the poleward flow of the Hadley cell due to eddy momentum fluxes (Schneider, 2006), and the boundary between the Hadley and Ferrel cells is shaped by these eddy momentum fluxes (Schneider, 2006; Lu et al., 2008; Ceppi and Hartmann, 2013; Choi et al., 2014). While the scaling theory can be adjusted to take into account the degree to which eddy fluxes draw the circulation away from angular momentum conservation (Kang and Lu, 2012), some bootstrap or input of the properties of the eddies is still needed to form a complete theoretical scaling for the Hadley cell width (Held, 2000).

Additionally, baroclinic instability is generally a feature of the eddy-driven jets, which can be well-separated from the subtropical jets at the edges of the Hadley cells. Despite the intermodel correlation between tropical expansion and increases in static stability, increases in static stability may not be the only process associated with tropical expansion. Instead, changes to the eddy phase speeds that lead to poleward shifts in the latitudes of wave breaking (Chen and Held, 2007) may be responsible for poleward shifts of the Hadley cell edges (Ceppi and Hartmann, 2013). Both occur simultaneously with increasing greenhouse gas concentrations and global-mean surface temperatures. It is therefore impossible to exclude other factors and conclude that the static stability increases alone drive tropical expansion.

Both Arctic warming and tropical upper-tropospheric warming scale with increases in global-mean surface temperature. Separating these influences on the tropical belt and any other feature of the climate system is not feasible in the experiments examined here and may not be possible in projections of future climate. Despite the significant variation in the magnitude of the model response to simple forcings, we find a robust physical scaling throughout the climate system, between the tropics and the poles and between the thermodynamics and the circulation.

*Acknowledgements.* We thank all participants of the Geoengineering Model Intercomparison Project and their model development teams, the CLIVAR/WCRP Working Group on Coupled Modeling for endorsing GeoMIP, and the scientists managing the Earth System Grid data nodes who have assisted with making GeoMIP output available. We also thank Ben Kravitz for supplying some model output. We acknowledge the World Climate Research Programme's Working Group on Coupled Modeling, which is responsible for CMIP, and we thank the climate modeling groups for producing and making available their model output. N. A. Davis was supported by a National Science Foundation Graduate Research Fellowship. The National Center for Atmospheric Research is supported by the National Science Foundation.



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



**Table 1.** The model name, modeling group or agency, the $4\times CO_2$ experiment top-of-atmosphere radiative forcing relative to piControl, and the G1 experiment residual top-of-atmosphere radiative forcing relative to the piControl experiment for each of the nine models examined. All radiative forcings are from Huneeus et al. (2014) and are in W/m$^2$. Information on the radiative forcings in the CSIRO-Mk3L-1-2 model is unavailable.

| Model | Group | $4\times CO_2$ radiative forcing (Wm$^{-2}$) | G1 radiative forcing (Wm$^{-2}$) |
|---|---|---|---|
| CanESM2 | Canadian Centre for Climate Modelling and Analysis | 8.0 | 0.0 |
| CCSM4 | National Center for Atmospheric Research | 6.2 | -0.5 |
| CSIRO-Mk3L-1-2 | Commonwealth Scientific and Industrial Research Organisation | N/A | N/A |
| GISS-E2-R | Goddard Institute for Space Studies | 7.8 | 1.4 |
| HadGEM2-ES | Met Office Hadley Centre for Climate Science and Services | 6.4 | 0.4 |
| IPSL-CM5A-MR | Institut Pierre Simon Laplace Climate Modelling Centre | 6.2 | 0.2 |
| MIROC-ESM | University of Tokyo, National Institute for Environmental Studies, and Japan Agency for Marine-Earth Science and Technology | 8.7 | 0.0 |
| MPI-ESM-LR | Max Planck Intitute für Meteorologie | 8.6 | 0.2 |
| NorESM1-M | Norwegian Climate Center | 6.8 | 0.4 |





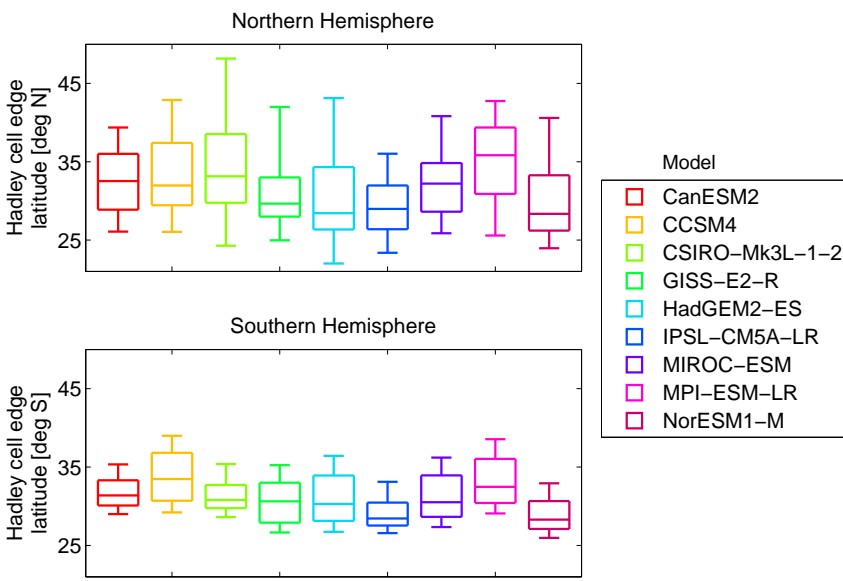

**Figure 1.** The piControl experiment climatology of the tropical belt edge latitudes for each of the nine models. The middle bar of each box represents the median and the top and bottom bars of each box represent the upper and lower quartiles, respectively, of the tropical belt edge latitudes. Whiskers indicate the maximum and minimum tropical belt edge latitude for the piControl experiment.

.





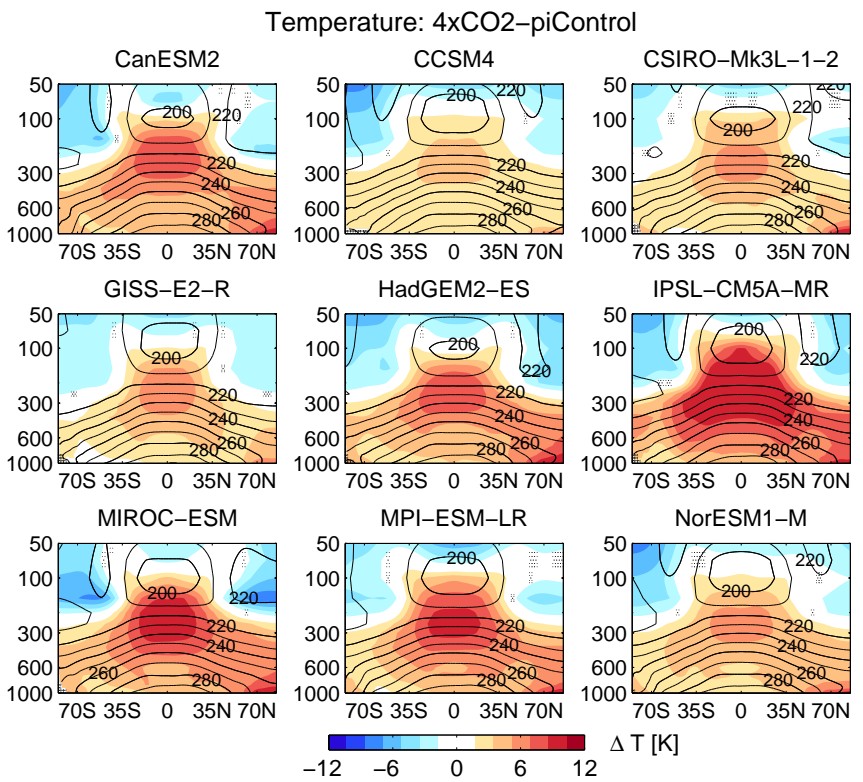

**Figure 2.** The difference in the zonal-mean temperature between the $4 \times CO_2$ and piControl experiments for each of the nine models. The $4 \times CO_2$ experiment temperature minus the piControl experiment temperature is shown in shading (Kelvin), while the piControl experiment temperature is shown by the black contours (Kelvin). Stippling indicates differences not significant at the 95% confidence level.





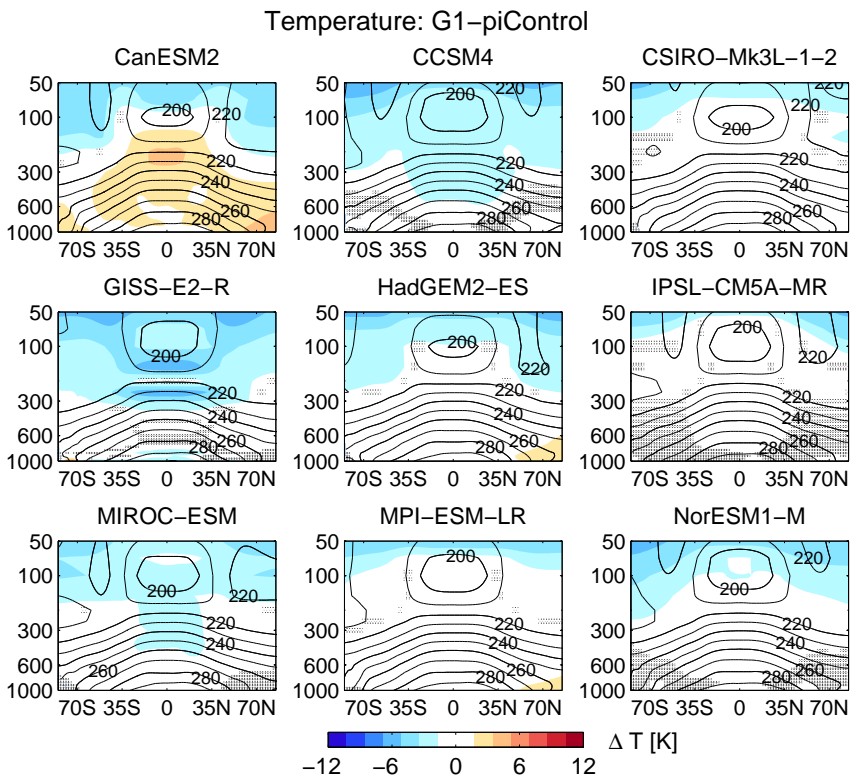

**Figure 3.** The difference in the zonal-mean temperature between the G1 and piControl experiments for each of the nine models. The G1 experiment temperature minus the piControl experiment temperature is shown in shading (Kelvin), while the piControl experiment temperature is shown by the black contours (Kelvin). Stippling indicates differences not significant at the 95% confidence level.




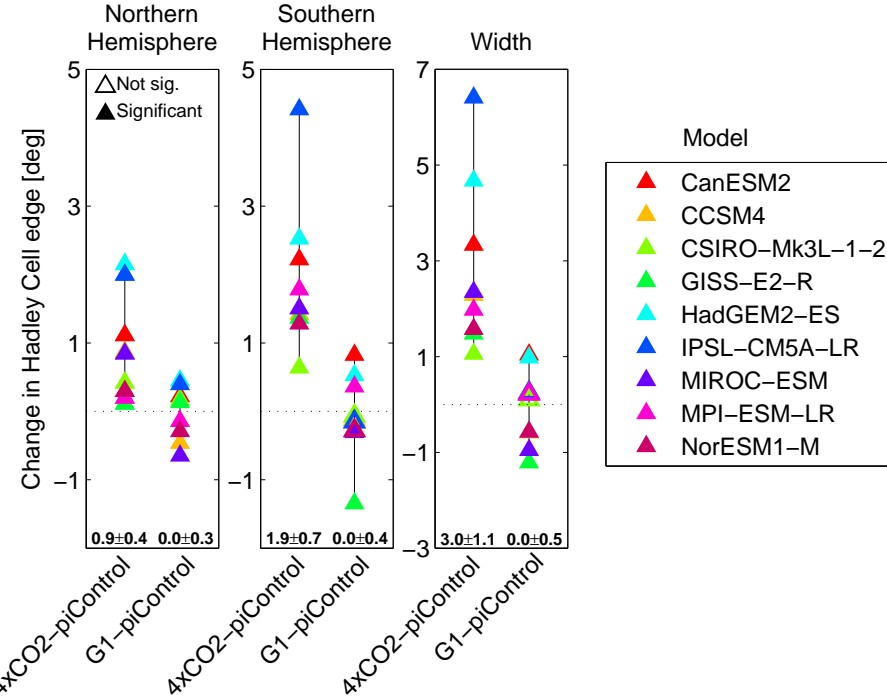

**Figure 4.** The change in the Hadley cell edge latitudes and width between the $4\times CO_2$ and piControl experiments and between the G1 and piControl experiments, for the Northern Hemisphere and Southern Hemisphere edge latitudes and for the total change in Hadley cell width (Width). Positive values indicate poleward expansion or an increase in width. Models with edge latitude or width changes significant at the 95% confidence level are indicated by solid symbols. The mean change in the tropical belt width or edge latitude and its 95% confidence interval in degrees latitude is shown at the bottom of each plot.





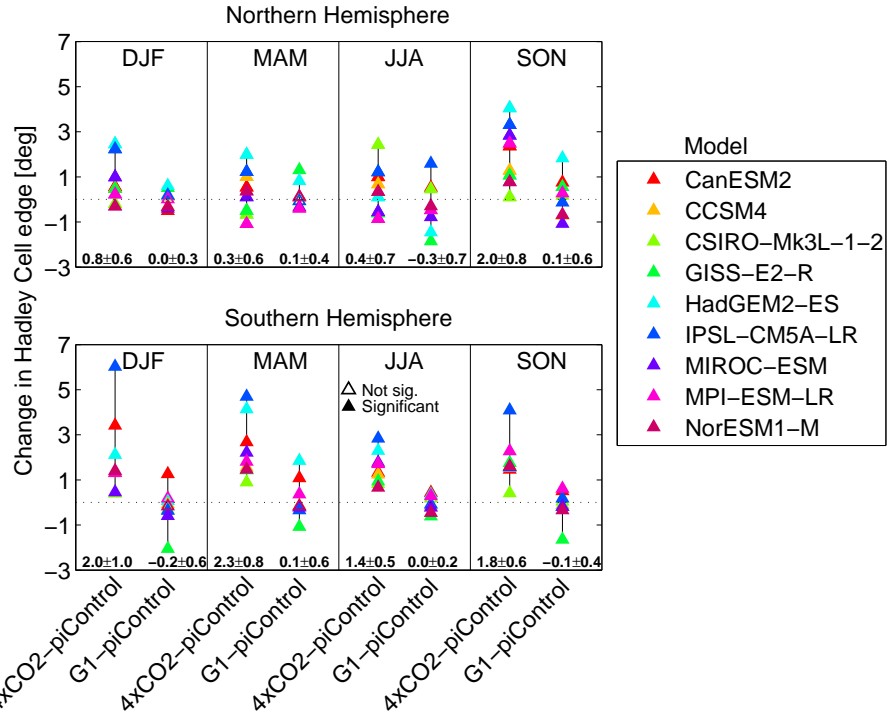

**Figure 5.** The seasonal change in the Hadley cell edge latitudes and width between the $4 \times CO_2$ and piControl experiments and between the G1 and piControl experiments, for the Northern Hemisphere and Southern Hemisphere edge latitudes. Positive values indicate poleward expansion. Models with edge latitude changes significant at the 95% confidence level are indicated by solid symbols. Values are shown for December through February (DJF), March through May (MAM), June through August (JJA), and September through November (SON). The mean change in the tropical belt width or edge latitude and its 95% confidence interval in degrees latitude is shown at the bottom of each plot.




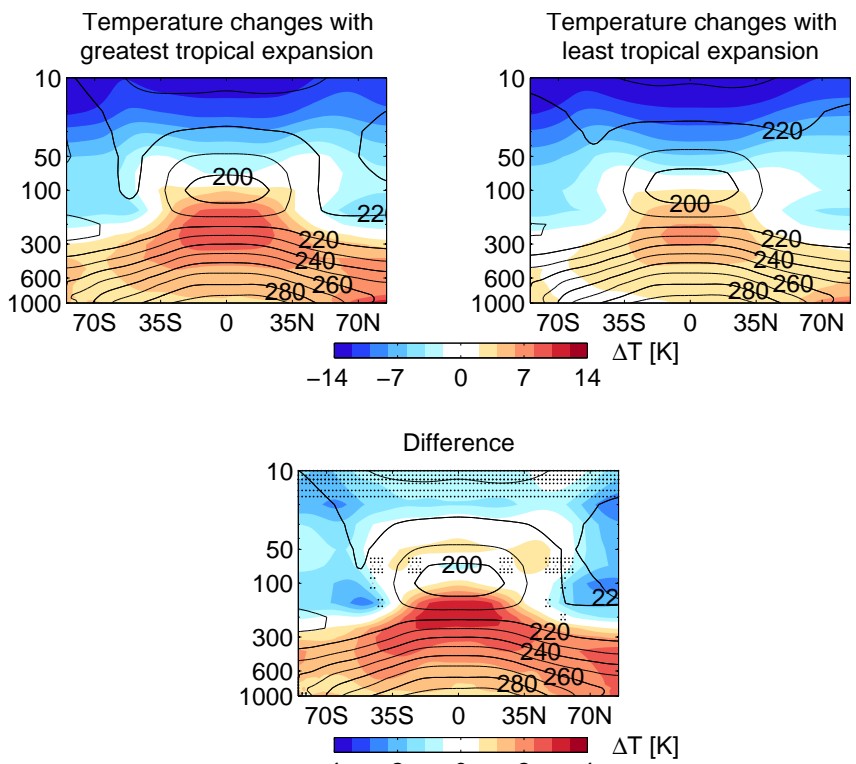

**Figure 6.** The difference in zonal-mean temperature between the $4\times CO_2$ and piControl experiments in the four models with the greatest tropical expansion (upper left) and in the four models with the least tropical expansion (upper right). The $4\times CO_2$ experiment minus the piControl experiment temperatures are shown in shading (Kelvin), while the piControl experiment temperatures are shown by the black contours (Kelvin). The difference in the $4\times CO_2$ experiment minus the piControl experiment temperatures between the models with the greatest and least tropical expansion is shown on the bottom, with shading indicating the difference (Kelvin) and black contours indicating the mean piControl experiment temperature (Kelvin) for all models. Stippling indicates changes not significant at the 95% confidence level.



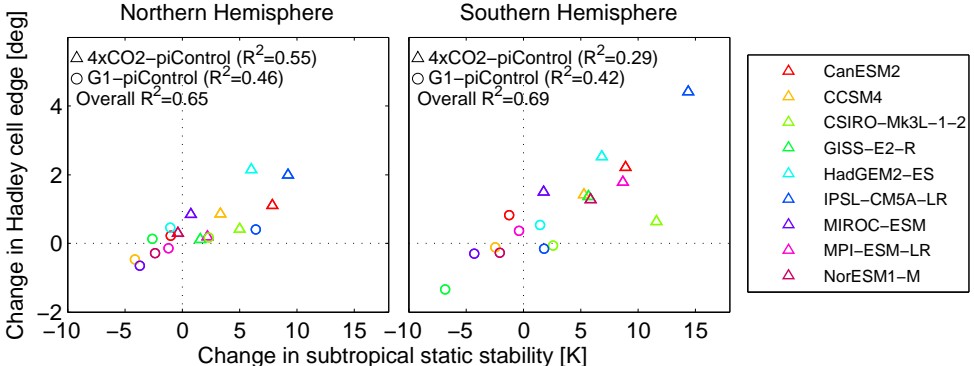

**Figure 7.** The change in the Hadley cell edge latitude versus the change in subtropical static stability in the Northern Hemisphere and in the Southern Hemisphere. For both hemispheres, positive changes in the Hadley cell edge latitude indicate poleward expansion. Shown are values for the $4 \times CO_2$ experiment minus the piControl experiment (triangles) and for the G1 experiment minus the piControl experiment (circles). The percent of the intermodel variation in the change in the Hadley cell edge latitude explained by the change in subtropical static stability between each experiment is indicated in each plot.

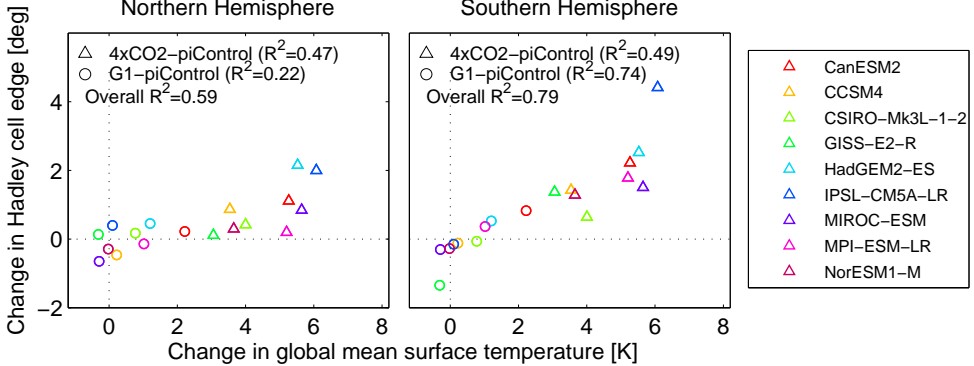

**Figure 8.** As in Fig. 7, but for the change in the Hadley cell edge latitude versus the change in global-mean surface temperature in the Northern Hemisphere and in the Southern Hemisphere.





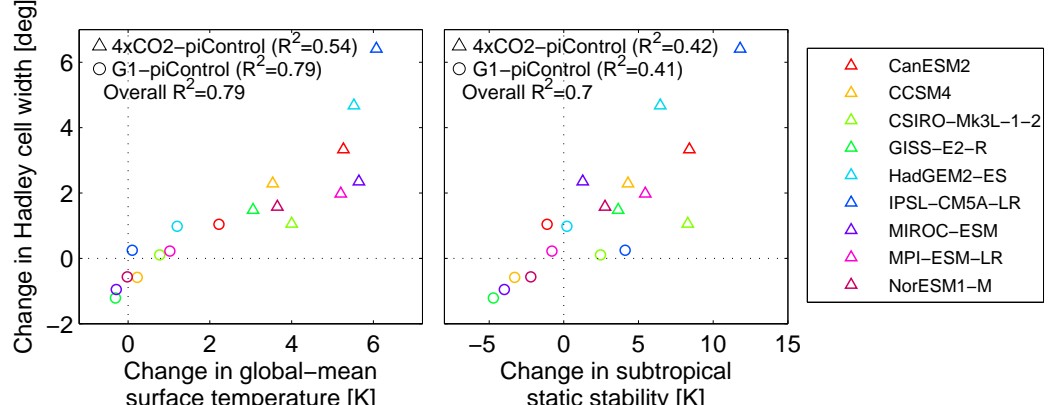

**Figure 9.** The change in the total Hadley cell width versus the change in global-mean surface temperature and the change in subtropical static stability. Positive changes in the Hadley cell width indicate tropical expansion. Shown are values for the $4\times CO_2$ experiment minus the piControl experiment (triangles) and for the G1 experiment minus the piControl experiment (circles). The percent of the intermodel variation in the change in the Hadley cell edge latitude explained by the change in global-mean surface temperature and the change in subtropical static stability between each experiment is indicated in each plot.

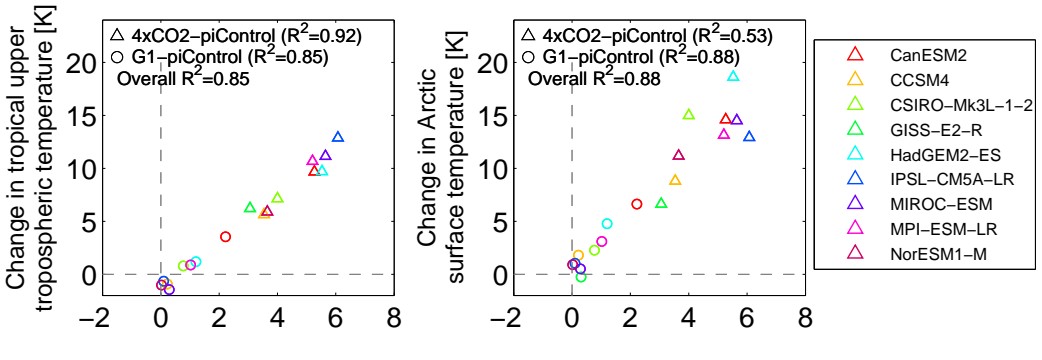

**Figure 10.** The change in tropical upper-tropospheric temperature versus the change in global-mean surface temperature (left), and the change in Arctic surface temperature versus the change in global-mean surface temperature (right), between the $4\times CO_2$ and piControl experiments (triangles) and between the G1 and piControl experiments (circles). Tropical upper-tropospheric temperature is defined as the mean temperature between 200 and 300 hPa and between 10S and 10N. Arctic temperature is defined as the mean surface temperature between 75N and 90N.