# Peer review of "Changes in the Width of the Tropical Belt due to Simple Radiative Forcing Changes in the GeoMIP Simulations"

_Atmospheric Chemistry and Physics, 2016_

## Referee Comment (RC1) · Anonymous Referee #1 · 20 May 2016

This study examines the response of the width of the tropical belt to an abruptly applied 4xCO2 forcing and an abruptly applied 4xCO2 forcing that is balanced by a decrease in the solar constant ("G1 experiment") in 9 CMIP5 models. The authors find that the tropical width responds unevenly to identical forcing across seasons and hemispheres. The response of the tropical width is correlated strongly with the response in global-mean surface temperature and the attendant increases in subtropical static stability, tropical upper tropospheric temperature, and Arctic surface temperature.

Overall, this paper is very well done. The text is written very clearly, and the figures are straightforward to interpret. What is particularly novel about this study is the usage of the GeoMIP experiments to demonstrate a linkage between tropical belt expansion and

global-mean surface temperature. My main criticism of this paper is that the authors fail to compare their results to a number of recent studies that have already examined simplified climate forcings in comprehensive global climate models, including the exact same abrupt 4xCO2 CMIP5 experiments that were examined here. The authors' assertions that "[no previous studies] have examined how comprehensive climate models respond to simplified climate forcings" (lines 8-9) and that "what is lacking is a study that applies simple climate forcings in clearly designed experiments to fully-coupled models" (lines 106-108) are too strong in my opinion. In many aspects, this paper is written more clearly and goes farther than previous studies, but I think it's important to put the new findings in much better context of previous work on the subject. Suggested revisions are detailed below.

Minor Revisions

GENERAL: As stated above, a greater cross-comparison of results with previous studies that used simplified climate forcings is warranted. A handful of these studies have already addressed the tropical expansion issue in some detail:

a) Polvani et al. (2011) force CAM3 with a (2000-1960) greenhouse gas forcing only and find a similar seasonality to the Southern Hemisphere Hadley cell edge response documented here (see their Fig. 13e).

b) McLandress et al. (2011) force CMAM with greenhouse gas forcing only and find no seasonality to the Southern Hemisphere Hadley cell edge response (see their Fig. 8).

c) Grise and Polvani (2014) use the abrupt 4xCO2 experiments from 23 CMIP5 models and find a strong correlation between the magnitude of Southern Hemisphere Hadley cell edge expansion and the global-mean surface temperature response during all seasons (similar to what is found here). A recent paper by the same authors addresses the influence of global-mean surface temperature warming on Northern Hemisphere Hadley cell edge expansion (Grise and Polvani 2016).

d) Vallis et al. (2015) use the 1%/year $CO_2$ increase runs from 35 CMIP5 models and find little correlation between global-mean surface temperature warming and the magnitude of Hadley cell expansion (see their Fig. 21).

Line 39: You might want to clarify here that the strength of the Hadley cell is actually projected to weaken in a warming atmosphere. (Vecchi and Soden 2007)

Line 137: I'm surprised that the circulation metrics adjust to the abrupt forcing in only two years. The point of this paper is that the Hadley cell edge responds to global-mean surface temperature warming, but the global-mean surface temperature warming continues throughout the duration of the 140-year run (as the ocean temperatures slowly warm). More could be said about this apparent contradiction.

Line 147: "are" is repeated twice.

Line 175: "Models with more equatorward edge latitudes in one hemisphere have more equatorward edge latitudes in the other hemisphere." It might be useful to provide the correlation value here.

Line 197: Could the non-uniform stratospheric cooling be due to variations in the strength of the Brewer-Dobson circulation, for example?

Lines 199-201: This is consistent with IPSL-CM5A-LR having one of the higher climate sensitivities of the nine models examined, and CCSM4 have one of the lowest. It might be useful to note somewhere on Figure 2 the climate sensitivities of the models.

Line 262-263: The lack of robustness in the Northern Hemisphere tropical expansion could reflect the compensating effects of two large robust responses, the effect of warming land on the tropical circulation and the effect of warming ocean on the tropical circulation (see Shaw and Voigt 2015).

Line 274: The upper stratospheric cooling appears to be similar in the two subsets of models. It's just the lower stratospheric cooling that varies.

Line 326: "The change in" is repeated.

Lines 389-402: Another potential mechanism to mention here is the upper tropospheric-lower stratospheric meridional temperature gradient. Certainly, increased subtropical static stability and increased tropical upper tropospheric temperatures go hand in hand. But, cooling in the polar lower stratosphere can shift the circulation poleward (e.g., Butler et al. 2010), and this has nothing to do with tropical heating or static stability. Both factors though change the meridional temperature gradient near the tropopause.

Table 1: Why are the radiative forcings listed in Table 1 different than those documented in Table 1 of Forster et al. (2013) for CMIP5 models (4xCO2)?

Figures 2 and 3: I believe that IPSL-CM5A-LR is mislabeled as IPSL-CM5A-MR.

Figure 6: Are these figures composited about the total width of the tropics (NH + SH)? If so, have you tried compositing about the NH and SH tropical edges separately? Are the results similar? Would you get the same composites if you subset the models by their global-mean surface temperature increase (instead of their Hadley cell widening)?

Figures 7-10: How do these relationships vary seasonally? Are the correlations uniform year-round, or do they have a distinct seasonality?

References:

Butler, A.H., D.W.J. Thompson, and R. Heikes (2010), The Steady-State Atmospheric Circulation Response to Climate Change-Like Thermal Forcings in a Simple General Circulation Model. J. Clim., 23, 3474-3496.

Forster, P. M., T. Andrews. P. Good, J. M. Gregory, L. S. Jackson, and M. Zelinka (2013), Evaluating adjusted forcing and model spread for historical and future scenarios in the CMIP5 generation of climate models, J. Geophys. Res. Atmos., 118, 1139–1150.
Grise, K. M., and L. M. Polvani (2014), Is climate sensitivity related to dynamical sensitivity? A Southern Hemisphere perspective, Geophys. Res. Lett., 41, 534–540.

Grise, K. M., and L. M. Polvani (2016), Is climate sensitivity related to dynamical sensitivity?, J. Geophys. Res. Atmos., 121, doi:10.1002/2015JD024687.

McLandress, C. et al. (2011) Separating the dynamical effects of climate change and ozone depletion: Part 2. Southern Hemisphere troposphere. J. Climate 24, 1850–1868.

Shaw, T. A., and A. Voigt (2015), Tug of war on summertime circulation between radiative forcing and sea surface warming, Nature Geosci., 8, 560-566, doi:10.1038/ngeo2449.

Vallis, G. K., P. Zurita-Gotor, C. Cairns, and J. Kidston (2015), Response of the large-scale structure of the atmosphere to global warming, Q. J. R. Meteorol. Soc., 141, 1479-1501, doi: 10.1002/qj.2456.

Vecchi, G. A., and B. J. Soden (2007), Global warming and the weakening of the tropical circulation, J. Clim., 20, 4316–4340.

---

## Referee Comment (RC2) · Anonymous Referee #2 · 1 Jun 2016

This paper documents the response of the width of the zonal mean tropical Hadley circulation to suddenly applied CO2 and solar forcings. The work is timely, the writing understandable, the methods appropriate, and the figures mostly clear. Some results worth highlighting include the following.

1. Reducing the solar constant to counteract greenhouse gas induced warming may maintain a steady Hadley circulation in spite of a cooling stratosphere.

2. Model dynamical sensitivity is distinct from climate sensitivity (see Grise & Polvani, 2016).

3. Well-mixed GHGs produces a seasonally varying shift.

My main criticism of the article is the same as RC1: the authors state that previous climate model studies have not "...examined how comprehensive climate models respond to simplified climate forcings." While this study is certainly useful, there is already other, similar work out there that ought to be discussed.

Specific comments

Line 102 - I don't believe the studies cited in this paragraph justify the statement that an increase in the height of the tropopause - independent from stratospheric cooling or tropospheric warming - drives a poleward shift in the circulation. I think this is an over-generalization.

Line 146 - What were some typical effective degrees of freedom calculated in this way?

Lines 237-245 - Good discussion of uncertainty.

Line 266 - "temperature structures" should probably be "zonal mean temperature structures"

Line 289 - I think that "successfully used to study tropical expansion" suggest more closure than the theory provides. It's proven useful but insufficient.

Lines 324-326 - Some clarification is needed here. I find the combination of "more linear", "more scattered," and "Despite the nonlinearity" all refer to the same result.

Technical comments

In the references There are missing DOIs (line 413), and several DOIs that point to the wrong paper (e.g. the DOI for the Allen & Sherwood reference about aerosols on lines 414-415 points instead to an Allen & Zender paper on Siberian snow cover).

The figures are nicely rendered, but some are carelessly produced. Figures 1, 4, 5, and 7-10 all use color as the only/primary way of conveying model information. "Do not use text color alone to convey information." I have attached a rasterized revision of Figure 1 which is much clearer, and a version of Figure 4 with a colorblind filter applied

(roughly 1 in 10 men will perceive the figures this way.) Use symbols, or just annotate points with model names where it matters.

[Figure]

Northern Hemisphere

Southern Hemisphere

**Fig. 1.** revised for clarity

[Figure]

**Fig. 2.** colorblind proof

---

## Author Response (AR1)

acp-2016-340: Changes in the Width of the Tropical Belt due to Simple Radiative Forcing Changes in the GeoMIP Simulations

**Response to RC1**

The authors thank Reviewer #1 for their time and their suggested revisions. Regarding major changes, we have updated figures so that they are color-blind-friendly and have added additional discussions of relevant previous work. We have also expanded the discussion of the seasonality of the width changes.

This study examines the response of the width of the tropical belt to an abruptly applied 4xCO2 forcing and an abruptly applied 4xCO2 forcing that is balanced by a decrease in the solar constant ("G1 experiment") in 9 CMIP5 models. The authors find that the tropical width responds unevenly to identical forcing across seasons and hemispheres. The response of the tropical width is correlated strongly with the response in global-mean surface temperature and the attendant increases in subtropical static stability, tropical upper tropospheric temperature, and Arctic surface temperature.

Overall, this paper is very well done. The text is written very clearly, and the figures are straightforward to interpret. What is particularly novel about this study is the usage of the GeoMIP experiments to demonstrate a linkage between tropical belt expansion and global-mean surface temperature. My main criticism of this paper is that the authors fail to compare their results to a number of recent studies that have already examined simplified climate forcings in comprehensive global climate models, including the exact same abrupt 4xCO2 CMIP5 experiments that were examined here. The authors' assertions that "[no previous studies] have examined how comprehensive climate models respond to simplified climate forcings" (lines 8-9) and that "what is lacking is a study that applies simple climate forcings in clearly designed experiments to fully-coupled models" (lines 106-108) are too strong in my opinion. In many aspects, this paper is written more clearly and goes farther than previous studies, but I think it's important to put the new findings in much better context of previous work on the subject. Suggested revisions are detailed below.

Minor Revisions

GENERAL: As stated above, a greater cross-comparison of results with previous studies that used simplified climate forcings is warranted. A handful of these studies have already addressed the tropical expansion issue in some detail:

a) Polvani et al. (2011) force CAM3 with a (2000-1960) greenhouse gas forcing only and find a similar seasonality to the Southern Hemisphere Hadley cell edge response documented here (see their Fig. 13e).
b) McLandress et al. (2011) force CMAM with greenhouse gas forcing only and find no seasonality to the Southern Hemisphere Hadley cell edge response (see their Fig. 8).
c) Grise and Polvani (2014) use the abrupt 4xCO2 experiments from 23 CMIP5 models and find a strong correlation between the magnitude of Southern Hemisphere Hadley cell edge

expansion and the global-mean surface temperature response during all seasons (similar to what is found here). A recent paper by the same authors addresses the influence of global-mean surface temperature warming on Northern Hemisphere Hadley cell edge expansion (Grise and Polvani 2016)

d) Vallis et al. (2015) use the 1%/year CO2 increase runs from 35 CMIP5 models and find little correlation between global-mean surface temperature warming and the magnitude of Hadley cell expansion (see their Fig. 21)

The authors thank the reviewer for these suggestions.

In response to this general comment, we agree this statement concerning idealized experiments in comprehensive models is too strong. The neglect of these papers was unintentional, and we thank the reviewer for listing these references. We have added a discussion of these papers so that our work is better situated in the context of previous work (see lines 108-117).

Line 39: You might want to clarify here that the strength of the Hadley cell is actually projected to weaken in a warming atmosphere. (Vecchi and Soden 2007)

This has been clarified by referencing Vecchi and Soden (2007) as well as Mitas and Clement (2006).

Line 137: I'm surprised that the circulation metrics adjust to the abrupt forcing in only two years. The point of this paper is that the Hadley cell edge responds to global-mean surface temperature warming, but the global-mean surface temperature warming continues throughout the duration of the 140-year run (as the ocean temperatures slowly warm). More could be said about this apparent contradiction.

Our analysis focused on the equilibrium response. Additionally, our discussion of the results in the initial submission did not argue for a mechanism of Hadley cell expansion but instead a consistent scaling across some climate parameters. We have clarified in the discussion of results that Hadley cell expansion and thermodynamic changes scale but only in the *equilibrium* and not *transient* sense, and that the timescale discrepancy rules out a direct thermodynamic mechanism (lines 399-404). We have also changed the title of the section investigating these correlations to "Intermodel differences in the tropical width response and associated thermodynamic changes".

Line 147: "are" is repeated twice.

Thank you, this has been fixed.

Line 175: "Models with more equatorward edge latitudes in one hemisphere have more equatorward edge latitudes in the other hemisphere." It might be useful to provide the correlation value here.

We have noted this ($R^2$=0.7, now on line 188).

Line 197: Could the non-uniform stratospheric cooling be due to variations in the strength of the Brewer-Dobson circulation, for example?

We have added a discussion noting this as a possibility (now on lines 207-209).

Lines 199-201: This is consistent with IPSL-CM5A-LR having one of the higher climate sensitivities of the nine models examined, and CCSM4 have one of the lowest. It might be useful to note somewhere on Figure 2 the climate sensitivities of the models.

Thank you for this suggestion, the equilibrium surface temperature responses have been added to each subplot for the 4xCO2 and G1 experiments.

Line 262-263: The lack of robustness in the Northern Hemisphere tropical expansion could reflect the compensating effects of two large robust responses, the effect of warming land on the tropical circulation and the effect of warming ocean on the tropical circulation (see Shaw and Voigt 2015).

Yes, this could certainly reflect these competing processes. This reference has been added to lines 274-277, thank you.

Line 274: The upper stratospheric cooling appears to be similar in the two subsets of models. It's just the lower stratospheric cooling that varies.

This is an interesting point that we had not appreciated – this certainly explains why the differences are also not significant. We have noted this in the text on lines 285-286.

Line 326: "The change in" is repeated.

Thank you, this is fixed.

Lines 389-402: Another potential mechanism to mention here is the upper tropospheric-lower stratospheric meridional temperature gradient. Certainly, increased subtropical static stability and increased tropical upper tropospheric temperatures go hand in hand. But, cooling in the polar lower stratosphere can shift the circulation poleward (e.g., Butler et al. 2010), and this has nothing to do with tropical heating or static stability. Both factors though change the meridional temperature gradient near the tropopause.

Yes, we agree and have noted this further possibility on lines 412-416.

Table 1: Why are the radiative forcings listed in Table 1 different than those documented in Table 1 of Forster et al. (2013) for CMIP5 models (4xCO2)?

These are the actual equilibrium radiative forcings for 4xCO2, whereas the table in Forster et al. (2013) displays the radiative forcings for a doubling of CO2 only. We use the values from Hunneus et al. so that the forcings from the G1 experiment can be directly compared.

If the forcings in Forster et al. are doubled they equal the forcings listed here and in Hunneus et al. 2014.

Figures 2 and 3: I believe that IPSL-CM5A-LR is mislabeled as IPSL-CM5A-MR.

Thank you, this is indeed in error.

Figure 6: Are these figures composited about the total width of the tropics (NH + SH)? If so, have you tried compositing about the NH and SH tropical edges separately? Are the results similar? Would you get the same composites if you subset the models by their global-mean surface temperature increase (instead of their Hadley cell widening)?

This is a good question. Yes, these are composited on the total change in width. There is not a substantial difference between the separate composites on Northern and Southern Hemisphere expansion, which is ultimately the reason we only show the composites on total width. However, there is slightly less dependence of the individual hemisphere's expansion on stratospheric cooling. We have noted this in the text on lines 286-288. For compositing on the change in global-mean surface temperature, the plots are essentially identical to Fig. 6 (this is probably apparent from Fig. 9).

Figures 7-10: How do these relationships vary seasonally? Are the correlations uniform year-round, or do they have a distinct seasonality?

There is indeed a seasonality to the correlations which we have not commented on. The existing discussion of seasonal expansion generally reflects the seasonality of the correlation between the change in global-mean surface temperature and seasonal expansion.

We have briefly noted some of the correlations in the text on lines 338-342. To summarize here, for the correlations between expansion and global-mean surface temperature, in the Southern Hemisphere the correlation is strongest in MAM ($R^2$=0.43), the season with the strongest mean expansion. A similar result is found for the Northern Hemisphere – the strongest correlation is in SON ($R^2$=0.31), the month with the strongest expansion. In the other seasons, there are no significant correlations between Northern Hemisphere expansion and the increase in global-mean surface temperature – though this could probably be inferred from Fig. 5. We have commented that these correlations generally reflect the strength and robustness of expansion in each season.

Tropical upper tropospheric warming has little seasonality. Arctic warming, on the other hand, is most correlated with both global-mean and tropical upper-tropospheric temperature changes in DJF ($R^2$~0.63), JJA ($R^2$~0.65), and SON ($R^2$~0.76). It is somewhat less correlated in MAM ($R^2$~0.56), though this is generally due to the CSIRO model, which is a significant outlier (it has far more warming in MAM compared to the other models, given its modest increase in surface temperature). The magnitude of Arctic warming is lowest in summer and highest in winter, consistent with previous research. For brevity we have only

noted that these indices are correlated with the change in global-mean surface temperature seasonally on line 352.

Going to finer timescales necessarily reduces the magnitude of the correlations. However, in general, models with a stronger response in one of these measures of climate have a stronger response in the others.

**acp-2016-340: Changes in the Width of the Tropical Belt due to Simple Radiative Forcing Changes in the GeoMIP Simulations**

**Response to RC2**

The authors thank Reviewer #2 for their time and their suggested revisions. Regarding major changes, we have updated figures so that they are color-blind-friendly and have added additional discussions of relevant previous work. We have also expanded the discussion of the seasonality of the width changes.

This paper documents the response of the width of the zonal mean tropical Hadley circulation to suddenly applied CO2 and solar forcings. The work is timely, the writing understandable, the methods appropriate, and the figures mostly clear. Some results worth highlighting include the following.

1. Reducing the solar constant to counteract greenhouse gas induced warming may maintain a steady Hadley circulation in spite of a cooling stratosphere.
2. Model dynamical sensitivity is distinct from climate sensitivity (see Grise & Polvani, 2016).
3. Well-mixed GHGs produces a seasonally varying shift.

My main criticism of the article is the same as RC1: the authors state that previous climate model studies have not "...examined how comprehensive climate models respond to simplified climate forcings." While this study is certainly useful, there is already other, similar work out there that ought to be discussed.

Thank you for this suggestion.

We agree this statement concerning idealized experiments in comprehensive models is too strong. The neglect of these papers was unintentional, and we have included the references suggested by RC1. We have added a discussion of these papers so that our work is better situated in the context of previous work (see lines 108-117).

Line 102 - I don't believe the studies cited in this paragraph justify the statement that an increase in the height of the tropopause - independent from stratospheric cooling or tropospheric warming - drives a poleward shift in the circulation. I think this is an over-generalization.

This is a fair point – tropopause height changes are indicative of other thermodynamic changes in the climate system, so they should not be discussed as independent factors. We have now made it clear that Lorenz and DeWeaver raised the tropopause height *and* cooled the stratosphere, and have removed mention of tropopause height as an independent mechanism for expansion (now line 104).

Line 146 - What were some typical effective degrees of freedom calculated in this way?

We have added the approximate degrees of freedom for the G1 (~400, shortest) and piControl (~4000, longest) experiments to line 157.

Line 266 - "temperature structures" should probably be "zonal mean temperature structures"

Thank you, we agree it is important to note that this study only focuses on the zonal-mean (now line 279).

Line 289 - I think that "successfully used to study tropical expansion" suggest more closure than the theory provides. It's proven useful but insufficient.

We agree that "successful" may give the impression that these scaling theories are in some way "proven". We have removed "successful". In the discussion on lines 399-404 we have also clarified that Hadley cell expansion appears to scale with the increase in static stability (and many other thermodynamic indices), but that actual mechanisms for expansion were not investigated here and are far from certain. Please also see the response to RC1 concerning the timescale of the adjustment to the radiative forcing, and the relationship between equilibrium climate and dynamical sensitivity.

Lines 324-326 - Some clarification is needed here. I find the combination of "more linear", "more scattered," and "Despite the nonlinearity" all refer to the same result

We have clarified the text in this section – now lines 335-337 and lines 343-345.

In the references There are missing DOIs (line 413), and several DOIs that point to the wrong paper (e.g. the DOI for the Allen & Sherwood reference about aerosols on lines 414-415 points instead to an Allen & Zender paper on Siberian snow cover).

Thank you, we have checked all DOIs and fixed any in error.

The figures are nicely rendered, but some are carelessly produced. Figures 1, 4, 5, and 7-10 all use color as the only/primary way of conveying model information. "Do not use text color alone to convey information." I have attached a rasterized revision of Figure 1 which is much clearer, and a version of Figure 4 with a colorblind filter applied (roughly 1 in 10 men will perceive the figures this way.) Use symbols, or just annotate points with model names where it matters.

We thank the reviewer for these suggestions and the example figures. We have changed Figure 1 to black and white and rotated it, as per the reviewer's suggestion, so that the differences between mean model edge latitudes are easier to discern. For Figures 4 and 5 we have changed the model identifiers to symbols (it is difficult to discern numbers on these plots), and for 7-10 we have changed the model identifiers to numbers (the symbols are difficult to discern in this case). In Figures 7-10, a black and gray scheme is used to

distinguish the different experiments and minimize any problems for readers with color-blindness. We appreciate these suggestions and will keep color-blind-friendly schemes in mind for future work.

**List of substantial changes**

Lines 41-42: added discussion of Hadley circulation weakening with global warming.

Lines 105-107: removed tropopause height as potential driver of Hadley cell expansion.

Lines 112-120: added discussion of relevant work that performed idealized experiments on fully-coupled climate models. The abstract was modified ("none" to "few") to reflect this, as well.

Lines 160-161: noted the approximate effective degrees of freedom for the experiments.

Line 191: added a correlation coefficient for the model-mean edge latitudes in the Northern and Southern Hemispheres.

Line 211: noted that Brewer-Dobson circulation changes could contribute to the structure of stratospheric cooling.

Lines 217-218: discussed the global-mean surface temperature response of the models with the strongest and weakest response.

Line 249: noted that this result agrees with the just-published manuscript Grise and Polvani (2016).

Lines 279-287: re-wrote this section to be less-confusing. It now simply focuses on the lack of robustness in Northern Hemisphere expansion and how this may be tied to the land/sea temperature contrast processes studied in Shaw and Voigt (2015).

Lines 295-298: added discussion of the lack of a difference in upper-stratospheric cooling between models with the greatest/least expansion. Also noted the result if one composites on individual hemisphere's expansion rather than on total with.

Section 4.1 title: changed to reflect the fact that we are not examining mechanisms.

Lines 334-335: Clarified that the result is robust over the domain of changes examined here.

Lines 351-356: re-wrote to be less-confusing. This paragraph now includes a discussion of the seasonality of the correlations, as well.

Lines 357-361: re-wrote to be less confusing – the word choice and order is more consistent now.

Lines 387-388: noted that some of these results reflect those of Grise and Polvani (2016).

Lines 418-423: added this discussion to make clear that we do not believe thermodynamic changes necessarily drive expansion, but merely that the equilibrium thermodynamic and dynamic sensitivities scale together.

Lines 430-434: added a discussion noting that stratospheric cooling can drive expansion, even though it may not effect static stability.

Throughout: changed "intermodel" to "inter-model".

Throughout: improved some grammar.

References:  fixed missing and incorrect DOI's.

Figure 1: Now black and white (more color-blind friendly) and vertically-oriented.

Figures 2,3: Global-mean surface temperature response is now included in each panel.

Figures 4,5: Now uses symbols and black/gray to denote models and statistical significance (more color-blind friendly). Using numbers (as Figures 7-10 now do) makes these plots unreadable.

Figures 7-10: Now uses numbers and black/gray to denote models and the two different experiments (more color-blind friendly). Using symbols (as Figures 4, 5 now do) makes these plots very difficult to read.

Figures 1-5,7-10: captions changed to reflect changes in the figures.

[revised manuscript text omitted]